# Factors related to time of stroke onset versus time of hospital arrival: A SITS registry-based study in an Egyptian stroke center

**Hossam M. Shokri**⬥*, **Nevine M. El Nahas, Hany M. Aref, Noha L. Dawood, Eman M. Abushady, Eman H. Abd Eldayem, Shady S. Georgy, Amr S. Zaki, Rady Y. Bedros, Mona M. Wahid El Din, Tamer M. Roushdy**

Department of Neurology, Faculty of Medicine, Ain Shams University, Cairo, Egypt

* hossam.shokri@med.asu.edu.eg

## Abstract

### Background

High-quality data on time of stroke onset and time of hospital arrival is required for proper evaluation of points of delay that might hinder access to medical care after the onset of stroke symptoms.

### Purpose

Based on (SITS Dataset) in Egyptian stroke patients, we aimed to explore factors related to time of onset versus time of hospital arrival for acute ischemic stroke (AIS).

### Material and methods

We included 1,450 AIS patients from two stroke centers of Ain Shams University, Cairo, Egypt. We divided the day to four quarters and evaluated relationship between different factors and time of stroke onset and time of hospital arrival. The factors included: age, sex, duration from stroke onset to hospital arrival, type of management, type of stroke (TOAST classification), National Institute of Health Stroke Scale (NIHSS) on admission and favorable outcome modified Rankin Scale (mRS $\leq$2).

### Results

Pre-hospital: highest stroke incidence was in the first and fourth quarters. There was no significant difference in the mean age, sex, type of stroke in relation to time of onset. NIHSS was significantly less in onset in third quarter of the day. Percentage of patients who received thrombolytic therapy was higher with onset in the first 2 quarters of the day (p = <0.001). In-hospital: there was no difference in percentage of patients who received thrombolytic therapy nor in outcome across 4 quarters of arrival to hospital.

**Data Availability Statement:** The data that support the findings of this study are available from the following link:

https://datadryad.org/stash/share/M2usEYIZ
bikcuU08ws_6ze_wnJ3NJPmyqqojJTse?Pg.

**Funding:** The author(s) received no specific
funding for this work.

**Competing interests:** The authors have declared
that no competing interests exist.

## Conclusion

Pre-hospital factors still need adjustment to improve percentage of thrombolysis, while in-hospital factors showed consistent performance.

## Introduction

Diseases of the circulatory system, including stroke are the main cause of mortality in Egypt [1], with a crude incidence rate (CIR) of 137,000 to 250,000 each year [2]. Over the last 4 years (2016–2020), stroke care in Egypt became in the center of attention of medical health providers. Intravenous recombinant tissue plasminogen activator (r TPA) was endorsed by ministry of health after a study that was conducted to assess the obstacles regarding its availability, and this resulted in 5 times increase in thrombolysis in Ain Shams University stroke centers [3].

Consequently, growing effort is dedicated to improve the outcome of patients who are presenting with acute ischemic stroke. 'Time is brain' in acute ischemic stroke management is translated into (onset to door and door to needle time) in order to administer the only approved treatment.

In 2018, Zakaria et al conducted a study in Egypt, to assess the causes of pre-hospital delay that would interfere with early arrival and administration of rTPA, and found that lack of patient or relative awareness about symptoms of stroke and waiting for symptoms to resolve spontaneously, in addition to lacking knowledge about the availability of acute intervention for stroke were the most common causes of prolonged onset to door [3]. Pre-hospital delay was estimated to range from 3–6 hours, and others reported that only 20% to 25% arrived to emergency within 3 hours of onset [4, 5].

The importance of door to needle time was assessed in a meta-analysis study that showed an increase of short-term mortality and disability at discharge in stroke patients presenting off-hours [6].

Previous studies reported lack of data documentation that resulted in difficulties in deducing time of stroke onset [4, 7], and recommended high quality data in order to evaluate and reduce causes of delayed access to hospital [5].

The aim of this registry-based study was to define factors related to time of stroke onset (day quarter, stroke subtypes, stroke severity, onset to door) and time of hospital arrival (door to needle) with the purpose of identifying those relevant to administration of thrombolytic therapy and to AIS outcome. The study has been conducted in the two certified stroke centers of Ain Shams University hospitals, both of which apply the acute stroke management protocol approved by the stroke chapter of the Egyptian Society of Neurology, Psychiatry, and Neurosurgery to be the standard protocol for Egyptian stroke units. And both have CT scan and MRI devices available over 24 hours. Thrombolytic therapy is available and covered either by medical insurance or by the reimbursement from the Ministry of Health.

## Methods

### Study design and population

This is a retrospective registry-based study derived from the Safe Implementation of Treatments in Stroke (SITS) of two stroke centers of Ain Shams University stroke centers, Cairo, Egypt, from January 2016 till December 2018, and after the approval of ethical committee of faculty of medicine, Ain Shams university. The two involved centers contribute to more than

50% of the data registered from Egypt, and are the only stroke centers in Egypt accredited by the LGA InterCert [8]. One of these centers is present in the teaching hospital and another in the specialized hospital, both of which are located in Eastern Cairo. Both centers serve the same catchment area of about a third of greater Cairo and its suburbs. Patients' data are routinely recorded including items relevant to time of stroke onset and hospital arrival. Only cases of ischemic stroke are included in analysis. In order to study the factors related to time of stroke onset and time of hospital arrival, the day was divided into 4 quarters: from 6:01 to 12:00 (1st quarter; morning), from 12:01 to 18:00 (2nd quarter; afternoon), from 18:01 to 24:00 (3rd quarter; evening), from 00:01 to 6:00 (4th quarter; night). Then factors are categorized according to its relevance into either (time of stroke onset) vs (time of hospital arrival).

Regarding time of stroke onset, we studied the difference in the following along the 4 quarters of the day: stroke frequency, age, sex, type of stroke according to TOAST classification [9], stroke severity at presentation measured by National Institute of Health Stroke Scale (NIHSS) [10], and onset of stroke to hospital arrival time (onset to door). Regarding the time of hospital arrival, we studied the difference in the following factors: door to needle and relative frequency of patients received thrombolytic therapy, in addition to favorable stroke outcome at 3 months defined as modified Rankin Scale ≤2 (mRS≤2) [11] to assess the hospital performance over the 4 quarters of the day.

## Statistical analysis

Statistical analysis was done using SPSS version 19th version Statistics (SPSS Inc., Chicago). The Shapiro-Wilks test was performed to test the normality of continuous data distribution. Median and interquartile range (IQR) were used for skewed data, whereas categorical data were presented as frequencies. Kruskal–Wallis test used to compare not normally distributed continuous variable with nominal independent variables. The chi-square test was used for comparison of nominal data.

## Results

### Demographic data

After excluding cases of hemorrhagic stroke, the studied population was 1450 of AIS cases as shown in flow chart (Fig 1). The median age was 63 (IQR 55–70 years), males accounted for 64.1%, and the most common vascular risk factors were hypertension (63.6%) followed by diabetes (48.5%), while the least risk factor was previous stroke within 3 months (1.8%). Onset to door ranged from 30 to 9900 min., with a median of 480 (IQR = 270–1380) min., baseline NIHSS ranged from 0 to 25, with a median of 7 (IQR = 4–11), out of all cases arriving to the 2 stroke centers 298 cases (20.6%) received thrombolytic therapy (Table 1).

### Factors related to time of stroke onset

Stroke onset had a higher frequency in the first quarter (morning) followed by the fourth (night) 40.8%, 28% respectively. There was no significant difference in the age, sex, or type of stroke in relation to time of stroke onset across the 4 quarters of the day. NIHSS was significantly less in patients with stroke onset in third quarter (p = 0.01). Type of stroke showed no variability in relation to time of onset (Fig 2). **Onset to door** was significantly delayed in patients with stroke onset in 3$^{rd}$ quarter (evening) (median 840; p <0.001) (Table 2). (Fig 3).

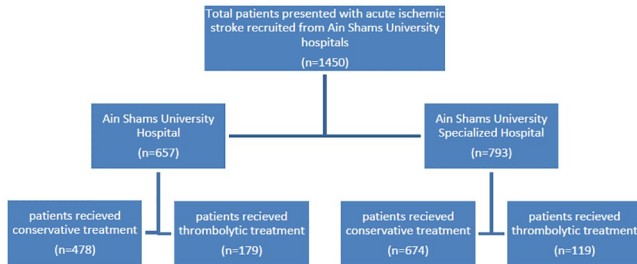

**Fig 1. Flowchart of total cases of acute ischemic stroke from SITS of Ain Shams University 2 stroke centers.**

**Table 1. Descriptive results and clinical characteristics of the studied subjects.**

| Variables | n = 1450 |
|---|---|
| *Age (median, IQR), years* | 63 (55–70) |
| *Male (%)* | 64.1% |
| *Hypertension (%)* | 63.6% |
| *Diabetes (%)* | 48.5% |
| *Dyslipidemia (%)* | 6.7% |
| *Smoking (%)* | 11.8% |
| *Previous stroke earlier than 3 months (%)* | 9.5% |
| *Previous stroke within 3 months (%)* | 1.8% |
| *Onset to door (median, IQR), min* | 480 (270–1380) |
| *Admission NIHSS (for 1351 patients only) (median, IQR)* | 7 (4–11) |
| *rtPA (%)* | 20.6% |

*IQR*: inter quartile range, *r TPA*: Intravenous recombinant tissue plasminogen activator.

## Factors related to time of hospital arrival

There was no difference in door to needle time nor in the percentage of patients who received thrombolytic therapy across 4 quarters (Figs 4 and 5). Also, the Outcome (mRS ≤2) showed no significant relation to time of arrival to hospital (missed patients are only 344 patients) (Table 3).

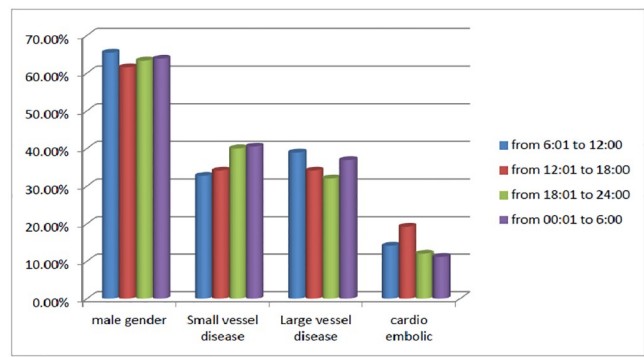

**Fig 2. Characteristics of patients according to time of stroke onset.**

**Table 2. Characteristics of patients according to time of stroke onset.**

|  |  | from 6:01 to 12:00 | from 12:01 to 18:00 | from 18:01 to 24:00 | from 00:01 to 6:00 | P VALUE |
|---|---|---|---|---|---|---|
| **Total** |  | 592 (40.8%) | 169 (11.7%) | 283(19.5%) | 406(28%) | 0.392 |
| **Age, median (IQR)** |  | 63 (55–70) | 64 (55–70) | 62 (55–69) | 63 (57–70) | 0.78 |
| **Male, %** |  | 65.4% | 61.5% | 63.3% | 63.8% | 0.8 |
| **Onset to door, median (IQR)** |  | 420 (180–1440) | 420 (150–1260) | 840 (480–2040) | 420 (330–720) | <**0.001**\* |
| **TOAST, %** | Small vessel disease | 32.7% | 34.1% | 40.0% | 40.5% | 0.1 |
|  | Large vessel disease | 38.9% | 34.1% | 32.0% | 36.9% |  |
|  | cardio embolic | 14.1% | 19.2% | 12.0% | 11.1% |  |
|  | cryptogenic | 11.9% | 11.4% | 13.1% | 8.3% |  |
|  | other determined | 2.2% | 1.2% | 2.5% | 2.8% |  |
|  | Sinus Venous Thrombosis | .2% | .0% | .4% | .5% |  |
| **Admission NIHSS (only 1351 patients), median (IQR)** |  | 7 (4–12) | 7 (5–10) | 6 (4–10) | 7 (4–10) | **0.01**\* |

\*p value is significant.

*IQR*: inter quartile range, *NIHSS*: National Institute of Health Stroke Scale (NIHSS).

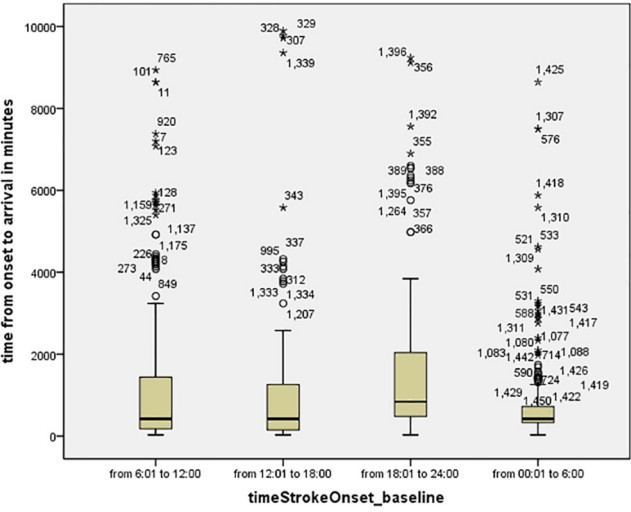

**Fig 3. Box plot of onset to door time along the 4 quarters of day according to time of stroke onset.**

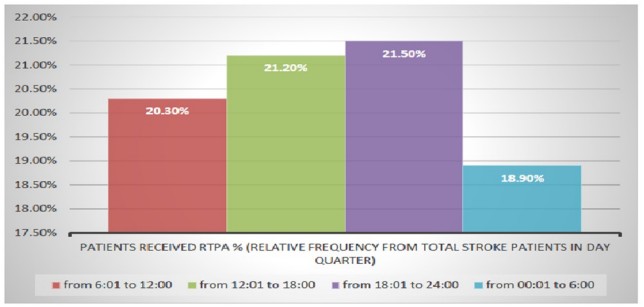

**Fig 4. Percentage of patients received rtPA along the 4 quarters of the day.**

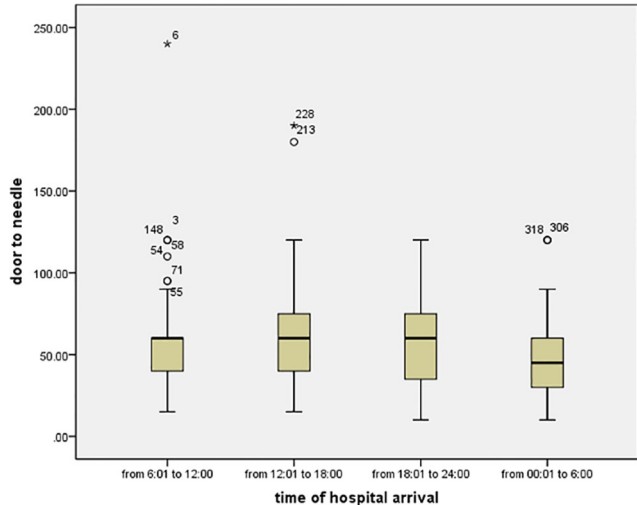

**Fig 5. Box plot of door to needle time along the 4 quarters of day according to time of hospital arrival.**

**Table 3. Characteristics of patients according to time of first hospital arrival.**

|  | from 6:01 to 12:00 | from 12:01 to 18:00 | from 18:01 to 24:00 | from 00:01 to 6:00 | P VALUE |
|---|---|---|---|---|---|
| **Door to needle, median, (IQR)** | 60 (40–60) | 60 (40–75) | 60 (30–75) | 45 (30–60) | 0.279 |
| **Patients received rtPA, %** | 20.3% | 21.2% | 21.5% | 18.9% | 0.93 |
| **Good outcome (mRS ≤2), %** | 69.0% | 73.2% | 71.9% | 71.7% | 0.62 |

*IQR*: inter quartile range.

N.B.: we only miss 344 patients on follow up the patients after 3 months for mRS.

## Discussion

In our study, we did not find significant predilection to age or sex in relation to time of stroke onset. Our finding was different from others who reported that onset during sleep was associated with younger age [7], and that males showed higher risk of large artery disease in the early morning [4].

Our stroke registry showed more frequent stroke onset in the first quarter of the day (morning), followed by the fourth quarter (night). Unfortunately, studies on the relationship of stroke to the time of the day showed relatively small number of patients and discordant results [4]. In spite of that most studies described the presence of circadian variation with a morning peak and another one sometimes reported in the afternoon [4, 7, 12–14].

Few studies found less incidence of stroke during sleeping hours but still higher in the morning than the afternoon [12, 13]. In line with this it was reported that all subtypes of stroke showed predominant onset in the morning [14, 15], while others showed two peaks one in the morning and another in the afternoon, with the least at night [7]. This circadian variation might be related to exogenous factors like sleep–awake cycles, physical activity, and up-right posture, and endogenous factors with their characteristic diurnal variation like blood pressure, autonomic system activity and hemostatic balance (with increased platelet agreeability, hyper-coagulability and hypo-fibrinolysis in the morning [16–18].

We found that type of stroke was not significantly related to time of onset; cases of small and large vessel strokes were more frequent than all other stroke subtypes across all quarters of the day. This can be attributed that those two types are in general the commonest types of stroke. Different results were described by Fodor et al. who found that cardio-embolic sub-group showed higher occurrence in sleep, while lacunar and cryptogenic strokes were more frequent during the first part of the morning, and large vessel disease occurred in the whole morning [7].

As for stroke severity, NIHSS was lower for onset in the third quarter (evening), denoting less stroke severity. Ripamonti et al. found that NIHSS was not related to stroke onset but onset during sleep was associated with bad mRS on discharge [4]. Similarly, Fodor et al. stated that night time onset of stroke showed less favorable outcome [7], this can be attributed to delayed hospital arrival for AIS during sleep, or to lack of intake of thrombolytic therapy during night shifts.

And despite that onset to door time was more prolonged with onset of stroke in the 3rd quarter (evening), yet the 3 months outcome did not differ from other quarters, possibly due to less stroke severity in the 3rd quarter as shown by a significantly lower NIHSS.

Onset to door time was mostly prolonged with onset of stroke in the 3rd quarter (evening), possibly related to a significantly less stroke severity as stated in the results. Factors that cause delay to treatment have been previously classified to either pre-hospital or in-hospital factors [19–25], however, time of onset and time of arrival at hospital were not among these factors.

In the current study, the percentage of patients who received thrombolysis was higher with onset in the first 2 quarters of the day and this is related to pre-hospital factors. But there was no difference in percentage of patients receiving thrombolytic therapy nor in outcome as measured by mRS whether hospital arrival was during morning working hours (first 2 quarters) or during night shifts (second 2 quarters). This might reflect a consistent in-hospital performance throughout the day. This could be explained by the availability of services in addition to a qualified well-trained medical team in both hospitals. It is worth saying that 24/7 availability of integrated stroke service was a prerequisite for accreditation of the stroke centers.

The study of the Swedish Stroke Registry noted that patients admitted in night time had a lower 30 and 90 day survival than those in daytime. That finding was explained by that door-to-needle time within 30 minutes was less likely during nighttime than daytime [26]. Interestingly, they found that university hospitals were more resilient to temporal variation- this could be applied to our study- than specialized non-university hospitals. Other studies also demonstrated an in-hospital delay of treatment attributable to night shifts and non-working hours [25, 27, 28].

## Conclusion

Time of stroke onset was not related to age, sex, type of stroke, only severity was less in stroke patients with evening onset that also showed delayed onset to door. Percentage of patients receiving thrombolytic therapy did not vary neither did the mRS with time of arrival to hospital. This might reflect a relatively consistence in-hospital performance at different times of the day.

### Strengths of the study

Our paper is one of few papers that tried to see the effect of time of stroke onset as well as time of hospital arrival on the severity, type of stroke, and outcome especially in Middle East. Moreover, the data of this study were derived from a well-structured registry where most of the

relevant data are regularly and routinely recorded and revised with a high degree of accuracy. Thus, fulfilling high quality data suggested by previous studies.

## Limitations of the study

Data about thrombectomy are missing due to the small number of thrombectomies done till the year 2018. The number increased starting from 2019 when it was reimbursed by the ministry of health.

mRS at 30 days is missing since we are committed to the SITS registry where only 3 months mRS is recorded.

Percentage of wake up stroke has not been reported in this study.

## Acknowledgments

This study was carried out in the both stroke units of Ain Shams University hospital and Ain Shams University Specialized hospital in Abbasia Square, Cairo, Egypt.

## Author Contributions

**Conceptualization:** Hossam M. Shokri, Nevine M. El Nahas, Hany M. Aref, Eman H. Abd Eldayem, Amr S. Zaki, Rady Y. Bedros, Mona M. Wahid El Din, Tamer M. Roushdy.

**Data curation:** Hossam M. Shokri, Nevine M. El Nahas, Hany M. Aref, Noha L. Dawood, Eman M. Abushady, Shady S. Georgy, Amr S. Zaki, Rady Y. Bedros, Tamer M. Roushdy.

**Formal analysis:** Hossam M. Shokri, Nevine M. El Nahas, Hany M. Aref, Eman H. Abd Eldayem, Shady S. Georgy, Tamer M. Roushdy.

**Investigation:** Hany M. Aref, Rady Y. Bedros, Tamer M. Roushdy.

**Methodology:** Hossam M. Shokri, Nevine M. El Nahas, Hany M. Aref, Eman H. Abd Eldayem, Shady S. Georgy, Rady Y. Bedros, Mona M. Wahid El Din, Tamer M. Roushdy.

**Project administration:** Hossam M. Shokri, Nevine M. El Nahas.

**Resources:** Shady S. Georgy.

**Software:** Hossam M. Shokri, Nevine M. El Nahas.

**Supervision:** Hossam M. Shokri, Hany M. Aref, Eman M. Abushady, Amr S. Zaki.

**Validation:** Nevine M. El Nahas, Hany M. Aref, Noha L. Dawood.

**Visualization:** Hossam M. Shokri, Nevine M. El Nahas, Hany M. Aref, Noha L. Dawood, Eman M. Abushady, Mona M. Wahid El Din, Tamer M. Roushdy.

**Writing – original draft:** Hossam M. Shokri, Nevine M. El Nahas, Eman H. Abd Eldayem, Amr S. Zaki, Mona M. Wahid El Din, Tamer M. Roushdy.

**Writing – review & editing:** Hossam M. Shokri, Nevine M. El Nahas, Tamer M. Roushdy.

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
