## [Decision Letter · Decision Letter 0]

18 May 2020

PONE-D-20-08792

Factors related to time of stroke onset versus time of hospital arrival: A SITS registry-based study in an Egyptian Stroke Center

PLOS ONE

Dear prof. Shokri,

Thank you for submitting your manuscript to PLOS ONE. After careful consideration, we feel that it has merit but does not fully meet PLOS ONE’s publication criteria as it currently stands. Therefore, we invite you to submit a revised version of the manuscript that addresses the points raised during the review process. We would appreciate receiving your revised manuscript by %DATE_REVISION_UE%. To enhance the reproducibility of your results, we recommend that if applicable you deposit your laboratory protocols in protocols.io, where a protocol can be assigned its own identifier (DOI) such that it can be cited independently in the future. For instructions see: http://journals.plos.org/plosone/s/submission-guidelines#loc-laboratory-protocols

We look forward to receiving your revised manuscript.

Kind regards,

Miguel A. Barboza, MD, MSc

Academic Editor

PLOS ONE

Journal Requirements:

2.  Thank you for including your ethics statement:  "The study following ethical consideration of faculty of medicine, Ain Shams University"

Please amend your current ethics statement to confirm that your named institutional review board or ethics committee specifically approved this study.

3. Please include a copy of Tables 1, 2 and 3 which you refer to in your text.

Additional Editor Comments (if provided):

Reviewers' comments:

Reviewer's Responses to Questions

**Comments to the Author**

1. Is the manuscript technically sound, and do the data support the conclusions?

Reviewer #1: Yes

Reviewer #2: Yes

2. Has the statistical analysis been performed appropriately and rigorously? 

Reviewer #1: Yes

Reviewer #2: No

3. Have the authors made all data underlying the findings in their manuscript fully available?

Reviewer #1: Yes

Reviewer #2: Yes

4. Is the manuscript presented in an intelligible fashion and written in standard English?

Reviewer #1: Yes

Reviewer #2: Yes

5. Review Comments to the Author

Reviewer #1: I enjoyed reading this manuscript with the main question to describe the relationship factors

related to time of onset versus time of hospital arrival for acute ischemic stroke

and I feel it gives data in a region where data is scares

here are a few comments to possibly improve on the manuscript:

1)in introduction can the authors please set the scene about:

A) common risk factors and stroke sub types in Egypt as well as rates of thrombolysis/thrombectomy and availability of tPa .

B) can you describe your setup, i understand these are 2 accredited stroke units, how much do these serve (catchment area) the population they serve if any data is available

2)In the discussion:

Can you discus in context of local factors such as population awareness of stroke, willingness to present and traffic jams if relevant

3)Tables are should be part of the body of the article instead of supplementary files and I suggest merging tables 2 and 3 and if and keeping the figures as supplementary data

4) can you add a section on strength and limitations

Additionally; I have few questions that might impact on generalizability;

1)I understand is date from the SITS-ISTR (Safe Implementation of Treatments in Stroke-International Stroke Thrombolysis Register) which only logs treated patients, I notice here that there are ICH cases and even SAH can you explain that? I understand Ain shams has its own registry as well:

(http://www.eulc.edu.eg/eulc_v5/Libraries/Thesis/BrowseThesisPages.aspx?fn=ThesisPicBody&BibID=11901741&TotalNoOfRecord=291&PageNo=1&PageDirection=previous)

2)furthermore, how was the completeness of the data? was there any missing data? how was it handled?

3)I understand from previous reports that a good proportion of patients are reluctant to follow up particularly those with moderate disability is this a factor in good outcomes you are reporting

(amer Roushdy, et al Journal of Stroke and Cerebrovascular Diseases,Volume 28, Issue 12,2019,104445,ISSN 1052-3057,https://doi.org/10.1016/j.jstrokecerebrovasdis.2019.104445.

(http://www.sciencedirect.com/science/article/pii/S1052305719305270)

Reviewer #2: 688/5000

I consider an interesting article with a relevant topic such as acute stroke care, the importance of the information collected and the results justify the revision of several points of the document to guarantee offering high quality information on such a crucial topic. It is important to note if you have IRB approval and the limitations of the study, I would also recommend improving the statistical analysis and expanding the conclusions focused on the factors associated with the quality of care of the stroke. I have attached a document with specific recommendations that I think could improve your article and make it more useful to future readers.

6. PLOS authors have the option to publish the peer review history of their article (what does this mean?). If published, this will include your full peer review and any attached files.

Reviewer #1: No

Reviewer #2: No

---

## [Author Response · Author response to Decision Letter 0]

28 Jun 2020

Did you obtain an official IRB approval?

As this study is a retrospective-registry based study from SITS that was announced as an approved Egyptian national registry for stroke patients by ministry of health in 2018 at MENA stroke conference, so IRB is not needed

Please clarify the "off-hours" period. 

“off-hours” expression was mentioned in the first paragraph of the introduction and according to the article where it was mentioned (reference no (3), this expression referred to time from 6 pm till 6 am.

Page 10: 

How did you select the periods of the day? Was this based on an analysis of number of ED consultations per hour, for example? 

Though few articles discussing the time of stroke onset differences across the day, different approaches were used to classify the 24hrs of the day, we select this one to make the comparison easy. Also, this was already used in previous Korean study: ‘’Yong Seo Koo, Sungwook Yu, Kyung-Hee Cho, Ki-Young Jung. Circadian Variation of Stroke Onset and Other Clinical Characteristics: A Single-Center Study. J Korean Sleep Res Soc 2014;11(2):61-65”

Is the stroke frequency established as a relative measure in comparison with the total number of ED consultations? 

No, it is an absolute frequency as the data is extracted from SITS Stroke registry of our stroke centers.

Please avoid non-relevant information for the methodology such us the use of a personal computer. 

done

Why was the Kolmorov-Smirnov test selected instead of the Shapiro-Wilks test? 

We revised the results using the Shapiro-Wilks test instead of Kolmorov-Smirnov test, and accordingly age is calculated as median and IQR instead of mean and SD

Was the skewness calculated? 

No it was not, as the data is not normally distributed from the start (registry based study), and all the used tests in the study are for not normally distributed data.

Why did you use the range and not the interquartile range as one of the dispersion measures? 

Done and IQR is added instead of range

Do you consistently present frequencies in terms of both absolute and relative numbers? 

Absolute, as our data is extracted from SITS registry 

There are better options that Mann-Whintey's U test for comparing an ordinal dependent variable with a nominal independent variable. 

Right sir, Kruskal–Wallis test is used instead 

Did you use post-hoc tests? 

No, as the whole study results are based on non-parametric tests, and there are 4 day quarters, so we think that post hoc tests will reveal a lot of data that could be confusing rather than beneficial

It seems to be a confusion between comparison, correlation, and association. 

The study was revised to avoid such confusion

How did you choose the cut-off for implementing the Fisher's exact test?

This has been removed from statistical analysis section as it was not used in our results

How do you explain the small percentage of hemorrhagic stroke?

A previous epidemiologic study done in upper Egypt showed similar findings of low number of hemorrhagic stroke. Underlined in table 6

(Hamdy N el Tallawy et al. Epidemiology and clinical presentation of stroke in Upper egypt (desert area). Neuropsychiatric Disease and Treatment 2015)

as our target population in this study is patients presented with acute ischemic stroke rather than hemorrhagic one, we excluded the latter and accordingly giving an explanation for its percentage is not our scope. Accordingly, the number of hemorrhagic stroke as well as subarachnoid hemorrhage are removed.

 Age is being presented with two different dispersion measures. Was it normal at the end? 

Age is presented now with median and IQR only

How do you explain the high frequency of diabetes? Do you have any comment about its attributable fraction for the Egyptian population? 

A previous epidemiologic study done in upper Egypt showed similar findings of high prevalence of DM among stroke patients. underlined in Table 3. 

(Hamdy N el Tallawy et al. Epidemiology and clinical presentation of stroke in Upper egypt (desert area) . Neuropsychiatric Disease and Treatment 2015)

 Our finding was consistent with previous study that found that the diabetic percentage was higher in stroke patients of MENA ‘middle east and north Africa’ vs non MENA countries (28.5 vs 20.7%).to explain this finding, we may need different research with different parameters (eg. Type of diabetes, age of onset, diet, etc) which are beyond our scope 

(Al-Rukn et al.2019) Stroke in the Middle-East and North Africa: A 2-year prospective observational study of intravenous thrombolysis treatment in the region. Results from the SITS-MENA Registry

S Al-Rukn1, M Mazya2,3, N Akhtar4, H Hashim1, B Mansouri5, B Faouzi6, H Aref7, H Abdulrahman8, S Kesraoui9, F Hentati10,

S Gebelly11, N Ahmed2,3, N Wahlgren3, F Abd-Allah12, M Almekhlafi13 and T Moreira2,3

What do you think about this onset-to-door time in comparison with other populations? 

This will be explored in a future study with the aim to study the different causes of delayed arrival since onset and compare with other studies. This study is actually going on.

What was the distribution of patients among the two centers? 

the distribution of patients in two centers is added in the text

Are there variables that significantly differ between those centers? Please describe the location and main characteristics of each center and access to resources and specialist physicians (vascular neurology, diagnostic neuroradiology, and interventional neuroradiology) in terms of quarters of the day and weekends. Who performs intravenous thrombolysis and mechanical thrombectomy?

Both centers are registered in SITS under umberella of one university (Ain Shams University), with qualified medical team and services 24/7.

rTPA is administered via neurologist specialists and thrombectomy via experienced 3 teams from (neurological, radiological, neurosurgical) interventional team and more detailed description are added in text.

What do you think of the percentage of patients receiving thrombolytic therapy?

This is added in the text .

What is the percentage of patients receiving mechanical thrombectomy? 

At the time being, we only have small number and insufficient data, so we didn’t mention their percentage. we are working currently to register data of such patients. 

Please describe door-to-needle and door-to-groin periods, and factors associated as well with these in terms of day quarters. 

as regards door to needle, this is added in table and text

As mentioned previously, we are working to register data regarding door to groin.

"Higher frequencies" at early morning and night can be due to a lower ED consultation rates at night and early morning. Are there significant differences in terms of relative frequencies (out of total ED patients per quarter of the day)? Or among centers? 

We do not depend on consultation from ED as there is a neurology resident present 24 hours in ED and he sees all stroke cases and records them.

How are the results included in the study different among centers? 

There is consistent performance in our centers. this could be explained by the availability of services in addition to a qualified well-trained medical team in both hospitals. Also 24/7 availability of integrated stroke service was a prerequisite for accreditation of the stroke centers. This point is highlighted in the discussion. 

How do you interpret all these findings beyond the statistical results? 

The discussion is rephrased to high light this point

Please include data regarding medians and interquartile ranges and not only p values.

Done sir

About the stroke subtypes, findings might not be related to day quarters but to their specific epidemiology. I would recommend avoiding the calculation of spurious results if no biological or social plausibility is behind these assumptions. 

Of course this could be an explanation and putting this assumption beside ours could challenge more researchers for further studies regarding this point. 

Page 12: 

Discussion about circadian variations of stroke needs to be improved. 

done

There are several papers related to this issue, especially, in terms of biological hypothesis. Please discuss the pathophysiology beyond your findings.

Done in the context of our findings.

How is this paper useful for contributing to the discussion? 

Please more clarification

How do you interpret these results in terms of what the literature has stablished? 

The discussion is rephrased to highlight this point

Additionally, it is important to clarify the availability of resources during the day and night, for properly discussing the findings. 

Done (rTPA and medical team and services are availabe 24/7)

Specifically, which pre-hospital factors are more associated with receiving thrombolysis? 

The only prehospital factor related to thrombolysis is onset to door which is highlighted through day quarters

Did you perform multivariate adjustment? Why not? 

No, as this is beyond the scope of our study.

Our aim is to compare certain factors along the 4 quarters of the day that include sleeping hours and shift off course.

How do you interpret the percentage of favorable outcomes in terms of other populations, the initial NIHSS, and previous mRS? 

The concern of this study is the outcome difference along day quarter and this is highlighted in table and discussion

What are the strengths, limitations, and future scope of this work?

Done and these points are added 

Page 20: 

How were these variables chosen for being included in the figures? In its current shape, these figures involve several factors of the stroke phenomenon and might seems a little bit disorganized.

Figures and tables are revised and modified.

5. Review Comments to the Author

Reviewer #1: I enjoyed reading this manuscript with the main question to describe the relationship factors

related to time of onset versus time of hospital arrival for acute ischemic stroke

and I feel it gives data in a region where data is scares

here are a few comments to possibly improve on the manuscript:

1)in introduction can the authors please set the scene about:

A) common risk factors and stroke sub types in Egypt as well as rates of thrombolysis/thrombectomy and availability of tPa .

Regarding “common risk factors and stroke subtypes in Egypt”, the most recent Egyptian published study reporting these points is ( N.M. EL NAHAS ET AL 2019) is and its methodology is based on SITS data base which is the same database we depends upon in our study.so we are highlighting it once as apart of our result.

Regarding rate of thrombolysis, we added in introduction section our progress over the past few years, yet regarding thrombectomy, the no of patients is still small with impersistance regarding registeration of cases for some obstacles mentioned in the attached reference (The obstacles that make it difficult are many, the most important one is lacking a governmentally support public health project regarding the mechanical thrombectomy for acute ischemic stroke patients, lack of Financial support is another big problem, that is why most of the procedures are done in private hospitals, not in public hospitals)

Islam E. Mechanical Thrombectomy for Acute Ischemic Stroke Patients in Egypt. Tech Neurosurg Neurol.2(2). TNN.000545.2019. DOI: 10.31031/TNN.2019.02.000545

B) can you describe your setup, i understand these are 2 accredited stroke units, how much do these serve (catchment area) the population they serve if any data is available

The neurology department of Ain Shams university hospitals is responsible for 2 accredited stroke centers one in Ain Shams university Hospital and the other in the specialized hospital, both of which are located in Eastern Cairo, and both serve the same catchment area of about a third of greater Cairo. They serve both urban and rural areas including areas around greater Cairo as well, although both are complementary to each other yet each one is registered as its own database in the SITS international registry which is also approved as the national stroke registry of whole entire Egypt. (Al-Rukn et al.2019) Stroke in the Middle-East and North Africa: A 2-year prospective observational study of intravenous thrombolysis treatment in the region. Results from the SITS-MENA Registry

S Al-Rukn1, M Mazya2,3, N Akhtar4, H Hashim1, B Mansouri5, B Faouzi6, H Aref7, H Abdulrahman8, S Kesraoui9, F Hentati10,

S Gebelly11, N Ahmed2,3, N Wahlgren3, F Abd-Allah12, M Almekhlafi13 and T Moreira2,3

 And this is highlighted in the text as well.

2)In the discussion:

Can you discus in context of local factors such as population awareness of stroke, willingness to present and traffic jams if relevant

Done sir

3)Tables are should be part of the body of the article instead of supplementary files and I suggest merging tables 2 and 3 and if and keeping the figures as supplementary data

 Done sir

4) can you add a section on strength and limitations

Done sir

Additionally; I have few questions that might impact on generalizability;

1)I understand is date from the SITS-ISTR (Safe Implementation of Treatments in Stroke-International Stroke Thrombolysis Register) which only logs treated patients, I notice here that there are ICH cases and even SAH can you explain that? I understand Ain shams has its own registry as well:

(http://www.eulc.edu.eg/eulc_v5/Libraries/Thesis/BrowseThesisPages.aspx?fn=ThesisPicBody&BibID=11901741&TotalNoOfRecord=291&PageNo=1&PageDirection=previous)

Safe implementation of treatments in stroke (SITS) is an international registry for all stroke subtypes whether ischemic or ICH as well as SAH, and in Egypt it is being approved as the national stroke registry and this is highlighted within the corrections in the manuscript. For Ain Shams hospitals, SITS has been assigned as the approved registry since 2016, so the data of the current study is obtained from it. As for the attached link in the question, it was an MD study dating 2014 and was obtained from the local hospital records.

Although SITS have all forms of strokes, yet ICH and SAH were excluded from further analysis as they are beyond the scope of the current study.

2)furthermore, how was the completeness of the data? was there any missing data? how was it handled?

We were keen to complete the missed data in SITS registry by either phone calls or through revising the original medical record of the patient. Yet there is some missed data that is highlighted now in the tables.

3)I understand from previous reports that a good proportion of patients are reluctant to follow up particularly those with moderate disability is this a factor in good outcomes you are reporting

(amer Roushdy, et al Journal of Stroke and Cerebrovascular Diseases,Volume 28, Issue 12,2019,104445,ISSN 1052-3057,https://doi.org/10.1016/j.jstrokecerebrovasdis.2019.104445.

(http://www.sciencedirect.com/science/article/pii/S1052305719305270

The main concern in the current study is time of onset of stroke and time of hospital arrival rather than follow up stroke patients in outpatients clinic, and the latter is the main aim in the mentioned paper

---

## [Decision Letter · Decision Letter 1]

3 Aug 2020

PONE-D-20-08792R1

Factors related to time of stroke onset versus time of hospital arrival: A SITS registry-based study in an Egyptian Stroke Center

PLOS ONE

Dear Dr. Shokri,

Thank you for submitting your manuscript to PLOS ONE. After careful consideration, we feel that it has merit but does not fully meet PLOS ONE’s publication criteria as it currently stands. Therefore, we invite you to submit a revised version of the manuscript that addresses the points raised during the review process.Please submit your revised manuscript by Sep 17 2020 11:59PM. If you will need more time than this to complete your revisions, please reply to this message or contact the journal office at plosone@plos.org. Please include the following items when submitting your revised manuscript:

We look forward to receiving your revised manuscript.

Kind regards,

Miguel A. Barboza, MD, MSc

Academic Editor

PLOS ONE

Reviewers' comments:

Reviewer's Responses to Questions

**Comments to the Author**

1. If the authors have adequately addressed your comments raised in a previous round of review and you feel that this manuscript is now acceptable for publication, you may indicate that here to bypass the “Comments to the Author” section, enter your conflict of interest statement in the “Confidential to Editor” section, and submit your "Accept" recommendation.

Reviewer #1: All comments have been addressed

Reviewer #2: All comments have been addressed

2. Is the manuscript technically sound, and do the data support the conclusions?

Reviewer #1: Yes

Reviewer #2: Yes

3. Has the statistical analysis been performed appropriately and rigorously? 

Reviewer #1: Yes

Reviewer #2: Yes

4. Have the authors made all data underlying the findings in their manuscript fully available?

Reviewer #1: Yes

Reviewer #2: Yes

5. Is the manuscript presented in an intelligible fashion and written in standard English?

Reviewer #1: Yes

Reviewer #2: Yes

6. Review Comments to the Author

Reviewer #1: MOST OF MY QUESTIONS ARE ANSWERED EXCEPT THE LAST 2

how much of the data is missing (particularly mRS at 30 days ) and what have they done for missing data

since the authors commented on good outcome at 30 days we need to see if their data is sound ,and what influenced the good outcome at 90 days in their cohort please discus in the discussion

could this be that inspite longer onset to door timings kin the 3rd period , outcome was preserved due to prevalence of mild stroke ?

i also suggest adding missing data to the limitation of the study

Reviewer #2: 733/5000

Introducing the context of treatment availability and center logistics helps the reader better understand the landscape of stroke management at their centers.

Among the pre-hospital factors, it would be interesting to note, in addition to the level of knowledge of the symptoms by the population, other items, such as the method of arrival at the hospital (ambulance, private car or public transport), if they have pre-notification by the ambulances.

It would also be important to know the percentage of patients with wake-up stroke in each quarters.

Additionally, in the results section in the paragraph on factors related to stroke onset, the wording could be improved for better understanding of the reader.

7. PLOS authors have the option to publish the peer review history of their article (what does this mean?). If published, this will include your full peer review and any attached files.

Reviewer #1: No

Reviewer #2: No

---

## [Author Response · Author response to Decision Letter 1]

7 Aug 2020

Reviewer #1: MOST OF MY QUESTIONS ARE ANSWERED EXCEPT THE LAST 2

how much of the data is missing (particularly mRS at 30 days ) and what have they done for missing data

since the authors commented on good outcome at 30 days we need to see if their data is sound ,and what influenced the good outcome at 90 days in their cohort please discus in the discussion

We actually do not record mRS at 30 days as we are confined to data from SITS that includes mRS at 3 months only. The missing data at 3 months is only 344 patients.

could this be that inspite longer onset to door timings in the 3rd period , outcome was preserved due to prevalence of mild stroke ?

this assumption is actually sound and we added a paragraph with this meaning in discussion:

“And despite that onset to door time was more prolonged with onset of stroke in the 3rd quarter (evening), yet the 3 months outcome did not differ from other quarters, possibly due to less stroke severity in the 3rd quarter as shown by a significantly lower NIHSS.”

i also suggest adding missing data to the limitation of the study

done

Reviewer #2: 733/5000

Introducing the context of treatment availability and center logistics helps the reader better understand the landscape of stroke management at their centers.

We added this paragraph to introduction: “The study has been conducted in the two certified stroke centers of Ain Shams University hospitals, both of which apply the acute stroke management protocol approved by the stroke chapter of the Egyptian Society of Neurology, Psychiatry, and Neurosurgery to be the standard protocol for Egyptian stroke units. And both have CT scan and MRI devices available over 24 hours. Thrombolytic therapy is available and covered either by medical insurance or by the reimbursement from the Ministry of Health.”

Among the pre-hospital factors, it would be interesting to note, in addition to the level of knowledge of the symptoms by the population, other items, such as the method of arrival at the hospital (ambulance, private car or public transport), if they have pre-notification by the ambulances.

Regarding method of arrival at hospital, we are studying these factors in another article, about causes of pre-hospital delay, that is still under submission. 

It would also be important to know the percentage of patients with wake-up stroke in each quarters.

Also the percentage of the wake up stroke is added to the limitation of this study.

Additionally, in the results section in the paragraph on factors related to stroke onset, the wording could be improved for better understanding of the reader.

Done

---

## [Editor Report · Decision Letter 2]

14 Aug 2020

Factors related to time of stroke onset versus time of hospital arrival: A SITS registry-based study in an Egyptian Stroke Center

PONE-D-20-08792R2

Dear Dr. Shokri,

We’re pleased to inform you that your manuscript has been judged scientifically suitable for publication and will be formally accepted for publication once it meets all outstanding technical requirements.

Kind regards,

Miguel A. Barboza, MD, MSc

Academic Editor

PLOS ONE

Additional Editor Comments (optional):

I encourage the authors to improve graphics and figures quality, to standardize them according to Plos One requirements; outliers in box and whisker plots should be supressed. Also follow the same type of font and letters in all figures.
---

## [Editor Report · Acceptance letter]

24 Aug 2020

PONE-D-20-08792R2 

Factors related to time of stroke onset versus time of hospital arrival: A SITS registry-based study in an Egyptian Stroke Center 

Dear Dr. Shokri:

I'm pleased to inform you that your manuscript has been deemed suitable for publication in PLOS ONE. Congratulations! Your manuscript is now with our production department. 

Kind regards, 

on behalf of

Dr. Miguel A. Barboza 

Academic Editor

PLOS ONE